# CAR T-Cell Targeting of Macrophage Colony-Stimulating Factor Receptor

**DOI:** 10.3390/cells11142190

**Published:** 2022-07-13

**Authors:** Daniela Yordanova Achkova, Richard Esmond Beatson, John Maher

**Affiliations:** 1CAR Mechanics Group, Guy’s Cancer Centre, School of Cancer and Pharmaceutical Sciences, King’s College London, Great Maze Pond, London SE1 9RT, UK; daniela.achkova@gmail.com (D.Y.A.); richard.beatson@ucl.ac.uk (R.E.B.); 2Department of Immunology, Eastbourne Hospital, Kings Drive, Eastbourne BN21 2UD, UK; 3Leucid Bio Ltd., Guy’s Hospital, Great Maze Pond, London SE1 9RT, UK

**Keywords:** chimeric antigen receptor, CAR T-cells, co-stimulation, cancer, immunotherapy, M-CSF, IL-34

## Abstract

Macrophage colony-stimulating factor receptor (M-CSFR) is found in cells of the mononuclear phagocyte lineage and is aberrantly expressed in a range of tumours, in addition to tumour-associated macrophages. Consequently, a variety of cancer therapies directed against M-CSFR are under development. We set out to engineer chimeric antigen receptors (CARs) that employ the natural ligands of this receptor, namely M-CSF or interleukin (IL)-34, to achieve specificity for M-CSFR-expressing target cells. Both M-CSF and IL-34 bind to overlapping regions of M-CSFR, although affinity of IL-34 is significantly greater than that of M-CSF. Matched second- and third-generation CARs targeted using M-CSF or IL-34 were expressed in human T-cells using the SFG retroviral vector. We found that both M-CSF- and IL-34-containing CARs enable T-cells to mediate selective destruction of tumour cells that express enforced or endogenous M-CSFR, accompanied by production of both IL-2 and interferon (IFN)-γ. Although they contain an additional co-stimulatory module, third-generation CARs did not outperform second-generation CARs. M-CSF-containing CARs mediated enhanced cytokine production and cytolytic activity compared to IL-34-containing CARs. These data demonstrate the feasibility of targeting M-CSFR using ligand-based CARs and raise the possibility that the low picomolar affinity of IL-34 for M-CSFR is detrimental to CAR function.

## 1. Introduction

The macrophage colony-stimulating factor receptor (M-CSFR, also known as the colony-stimulating factor receptor) is encoded by the c-*fms* proto-oncogene and is widely expressed throughout the mononuclear phagocytic lineage [1,2]. Aberrant expression of M-CSFR is prevalent in a range of cancers, including breast, ovarian, pancreatic and prostate carcinoma, malignant mesothelioma and both anaplastic large-cell and Hodgkin’s lymphoma [3]. Moreover, M-CSFR is highly expressed on immunosuppressive M2-polarised macrophages which are associated with poor outcome when present in the tumour microenvironment [4,5]. Accordingly, a number of monoclonal antibodies and small-molecule inhibitors of this receptor are under development for cancer therapy [6]. Despite their propensity to reduce circulating monocyte numbers, these agents are generally well-tolerated.

In recent years, cellular immunotherapies have emerged to complement traditional drug treatments for cancer. Chimeric antigen receptors (CARs) are synthetic fusion proteins that can be used to redirect the specificity of polyclonal T-cells or other immune cell populations against a native tumour-associated cell surface target [7]. In first-generation (1G) CARs, the endodomain consists of an activation module alone (e.g., CD3ζ), but this design proved ineffective when evaluated clinically [8]. Second-generation (2G) CARs also contain a single co-stimulatory domain, placed upstream of the activation module [9]. Immunotherapy using CD19-specific 2G CAR T-cells that contain either CD28 or 4-1BB has transformed the management of relapsed/refractory B-cell malignancy [10]. Similarly, 2G CARs targeted against B-cell maturation antigen have achieved highly impressive results in patients with multiple myeloma [11]. Third-generation (3G) CARs contain two co-stimulatory units in addition to an activation domain [12]. Recently, single-chain antibody fragment (scFv)-containing 3G CARs have been engineered to target M-CSFR and have achieved both tumour cytolytic activity and cytokine production. However, this was only reported with target cells in which M-CSFR expression was enforced by gene transfer [13].

Since some scFv-based CARs are susceptible to tonic signalling [14], we set out to engineer 2G and 3G CARs in which specificity for M-CSFR is conferred by either of the two natural ligands of this receptor. Macrophage colony-stimulating factor (M-CSF) and interleukin (IL)-34 are homodimeric ligands that promote M-CSFR dimerisation, binding with intermediate (K_d_ 34 pM) and low picomolar affinity (K_d_ 1 pM), respectively [15]. As a result, both M-CSF and IL-34 promote the survival, proliferation and differentiation of cells of the mononuclear leukocyte lineage [16]. Although these two cytokines display limited protein sequence homology, they share structural features and bind competitively to the extracellular domain (D)2 and D3 immunoglobulin-like loops within the M-CSFR [17]. Using these binding moieties, we sought to evaluate the feasibility of targeting tumour-associated M-CSFR using 2G and 3G CARs.

## 2. Materials and Methods

### 2.1. Cell Lines

All cell lines used in this study were validated by STR typing and were confirmed as mycoplasma-negative. The K299, DEL, FE-PD, JB6 (all derived from anaplastic large-cell lymphoma), KM-H2 and L540 (both derived from Hodgkin’s disease) cell lines were provided by Professor Stephan Mathas (Max-Delbrück Center for Molecular Medicine, Germany). The T47D breast cancer cell line was a gift from Professor Joy Burchell, King’s College London, United Kingdom. Cells were maintained in RPMI or DMEM medium that contained 10% FBS and GlutaMax (termed R10 or D10 medium, respectively). PG13 retroviral packaging cells were obtained from the European Collection of Cell Cultures (ECACC). 293VEC-RD114^TM^ retroviral packaging cells were a gift from Dr Manuel Caruso (Centre de recherche du CHU de Québec, Canada). Both retroviral packaging cell lines were maintained in D10 medium. Cell lines were engineered by retroviral transduction using the SFG vector to express RFP/ffLuc [18] or human *FMS* [19] (encodes for human M-CSFR), where indicated.

### 2.2. Human Samples

Blood samples were collected from healthy volunteers (male and female, aged 18–65 years). Approval was granted by a National Health Service Research Ethics Committee (code 09/H0804/92).

### 2.3. Retroviral Constructs

To generate M-CSF-targeted CARs, an NcoI/NotI flanked synthetic cDNA was generated (Genscript, Hong Kong, China) in which a human CD8α leader peptide was fused to human M-CSF isoform 3 (codons 33–189). This cDNA fragment was substituted for the small NcoI/NotI fragment in the previously described SFG T1E28z or SFG T1NA retroviral plasmids [20], thereby generating CARs designated M-2 and M-Tr. M-3 and all IL-34-targeted CARs (34-2, 34-3 and 34-Tr) were constructed using polymerase incomplete primer extension cloning, employing IL-34 isoform 1 (codons 1–242) as a targeting moiety in the latter. Structures of all CARs are illustrated in Figure 1A. All CARs were stoichiometrically expressed in human T-cells using a T2A ribosomal skip peptide and RRKR furin cleavage site alongside a chimeric cytokine receptor designated 4αβ (Figure 1B). The purpose of the furin cleavage site was to remove the peptide overhang left by the T2A peptide on the upstream polypeptide. 4αβ provides a selective IL-2/15 signal upon binding of IL-4 [21]. In each case, the 4αβ cDNA was inserted into the unique NcoI restriction site within the vector, thereby placing this insert upstream of the CAR cDNA.

To monitor tumour cell viability using luciferase assays, ffLuc was expressed in target cells by retroviral transduction [18]. The SFG *FMS* construct which encodes M-CSFR was previously described [19].

### 2.4. Transduction and Expansion of Human T-Cells

Retroviral vector was generated from PG13 or 293VEC-RD114^TM^ retroviral packaging cells [22]. Methods used to achieve retroviral transduction of activated human T-cells have been described previously [23,24].

### 2.5. FACS Analysis

All incubations were performed on ice. To detect M-CSF-containing CAR expression, cells were incubated with 500 ng/μL of goat anti-human M-CSF (Sigma-Aldrich, Poole, UK). Bound antibody was detected using FITC-conjugated anti-goat Ig 1 μg/μL (DAKO, Stockport, UK). To detect IL-34-containing CARs, cells were stained with anti-human IL-34 (clone 1D12) (Abcam, Cambridge, UK). Bound antibody was detected using PE-conjugated goat anti-mouse IgG (Invitrogen, Horsham, UK). The 4αβ chimeric cytokine receptor was detected using anti-human CD124 PE-conjugated (BD Biosciences, Wokingham, UK). Expression of human M-CSFR was detected using anti-human M-CSFR (clone 3-4A4) (Santa-Cruz, Heidelberg, Germany). Bound antibody was detected using PE-conjugated goat anti-rat IgG (Invitrogen). As a control, monocytes were isolated from PBMC using magnetic microbead selection for CD14^+^ cells (Miltenyi Biotec, Berdisch Gladbach, Germany) and differentiated over 7 days to either M1-polarised macrophages (using 20 ng/mL of recombinant human GM-CSF; Peprotech, London, UK) or M2-polarised macrophages (using 50 ng/mL of recombinant human M-CSF, which was kindly provided by Dr Nicholas Dibb, Imperial College London, UK). Flow cytometry was undertaken with a FACSCanto II or LSR Fortessa (BD Biosciences) using FACSDiva software (BD Biosciences). Collected data were analysed using FlowJo, LLC (FlowJo, Ashland, OR, USA).

### 2.6. Enzyme-Linked Immunosorbent Assay

Tumour cells were plated at a density of 0.8 × 10^6^ cells/mL. After 48 h, medium was collected and analysed for M-CSF and IL-34 concentrations using a Duoset ELISA kit (R&D Systems, Abingdon, UK). Reagent diluent and medium were used as negative controls.

Media that were collected from CAR T-cell/tumour cell co-cultures were analysed for IFN-γ or IL-2 using ELISA (eBioscience, San Diego, CA, USA), as described by the manufacturers.

### 2.7. Western Blotting

T-cell lysates were generated in RIPA lysis buffer, comprising 150 mM of NaCl, 1% NP-40, 1% sodium deoxycholate, 0.1% SDS and 50 mM of Tris HCl, pH 7.5, supplemented with 1× cOmplete^TM^ Protease inhibitor combination (Sigma-Aldrich). Lysates were centrifuged at 16,200× *g* and the pellet was discarded. Supernatant was mixed 3:1 with the protein-loading buffer (50 mM of Tris HCl, 100 mM of DTT, 2% (*w/v*) SDS, 0.1% (*w/v*) bromophenol blue, 10% (*v/v*) glycerol) and was heated to 90 °C for 10 min. After a further centrifugation step at 16,200× *g*, lysates were injected into appropriate wells of a pre-casted polyacrylamide gel. Electrophoresis was performed for 2 h at 20 V/cm. Protein transfer to a nitrocellulose membrane was undertaken using the Mini Trans-Blot system (Bio-Rad, Hercules, CA, USA), which was run overnight at 30 V/90 mA while maintaining a temperature of 4 °C. The nitrocellulose membrane was then incubated in blocking buffer (TBS, 5% (*w/v*) skimmed milk powder) for 1 hour at room temperature to prevent non-specific antibody binding. Next, anti-human CD3ζ antibody (BD Biosciences) was added at the manufacturer’s recommended concentration in blocking buffer, and this was added to the membrane for 60 min. The membrane was next washed for 10 min in TBST (50 mM of Tris, 150 mM of NaCl, 0.05% Tween-20) and this washing procedure was repeated a total of three times. A horse radish peroxidase (HRP)-conjugated goat anti-mouse secondary antibody (Dako) was diluted in blocking buffer to the recommended concentration and incubated for 60 min at room temperature with the membrane. After three further TBST washes, a pre-mixed ECL solution was added to the nitrocellulose membrane. Excess ECL solution was drained from the membrane after a 60 s incubation period. Next, the membrane was carefully covered in Saran wrap and placed within a film cassette. Working in the dark, a chemiluminescent hyper-film was placed on top of the covered membrane. After the desired exposure time, the film was developed, a procedure undertaken with a Jet Optimax X-ray developer (Optimax, London, UK).

### 2.8. Killing Assays

To assess cytolytic activity of CAR T-cells, tumour and lymphoma cells were co-cultivated with T-cells at a one-to-one effector to target (E:T) ratio. In the case of T47D, tumour cell viability was determined using an MTT assay, as described in [25]. In the case of M-CSFR-expressing lymphoma cells, viability was determined by luciferase assays, as described in [26]. Using either assay, percentage of target cell viability was calculated using the following formula: (absorbance/luminescence of tumour cells cultured with T cells divided by absorbance/luminescence of tumour cells alone) × 100%.

### 2.9. Statistical Analysis

All statistical testing was undertaken using GraphPad Prism (version 9.4.0, GraphPad Software, San Diego, CA, USA). When analysing datasets derived from multiple groups, one-way or two-way ANOVA testing was performed. Post hoc analysis was undertaken using Tukey’s multiple comparisons test. Correlation testing was performed using simple linear regression analysis.

## 3. Results

### 3.1. Engineering of CAR T-Cells Targeted against Macrophage Colony-Stimulating Factor Receptor

Human M-CSF and IL-34 were used to engineer 2G (CD28 + CD3ζ) and 3G (CD28 + 4-1BB + CD3ζ) CARs with specificity for M-CSFR. These were dubbed M-2, M-3, 34-2 and 34-3, respectively, and were compared to CD3ζ endodomain truncated (Tr) signalling null controls (M-Tr and 34-Tr, respectively; Figure 1A). To ensure comparable receptor expression levels across different experiments, each CAR was co-expressed with a chimeric cytokine receptor designated 4αβ, in which the human IL-4Rα ectodomain has been joined to the transmembrane/endodomain of IL-2/15Rβ (Figure 1B). 4αβ mediates enrichment of transduced cells upon culture in IL-4 as the sole cytokine support. Importantly, anti-tumour activity and type 1 polarity of the expanded T-cells is fully maintained [21]. Stoichiometric transgene co-expression was achieved using a T2A-containing SFG retroviral vector (Figure 1C).

Stable retroviral packaging cell lines were generated to engineer human T-cells to express each of these vectors. Expression of each transgene was demonstrated on retroviral packaging cells by flow cytometry (Figure 2A). Following transduction of human T-cells, Western blotting analysis was performed under reducing conditions to investigate if CARs expressed at the predicted molecular weights (mw) of 45.6 kDa (M-2), 50.5 kDa (M-3), 52.9 kDa (34-2) and 58.2 kDa (34-3) (Figure 2B). The blot was probed with anti-CD3ζ antibody, which allows detection of CAR band(s) together with endogenous CD3ζ. CAR and presumed degradation (Deg.) bands are indicated on the blot. Duplicate lanes are shown for M-2 and 34-2 (second lane indicated by an asterisk). In the case of M-2* and 34-2*, CARs were fused to a C-terminal T2A peptide, which accounts for the slightly increased molecular mass in each case. Two CAR bands were noted in the case of IL-34 CARs, perhaps owing to its propensity to undergo C-terminal glycosylation [27]. The endogenous CD3ζ band was seen at similar intensity in all lanes, confirming equal protein loading of each lane. Transduced T-cells were enriched by culture in IL-4, achieving comparable CAR expression across groups (Figure 2C). Representative examples of cell surface CAR expression are shown in Figure 2D.

### 3.2. M-CSF-Targeted CAR T-Cells Mediate Enhanced Cytokine Production When Cultured with T47D FMS Tumor Cells

To perform an initial comparison of the function of the CARs described above, T47D cells were selected as an M-CSFR negative control tumour cell line [28]. To engineer high-level M-CSFR expression in these cells, they were transduced with the SFG *FMS* retrovirus. Confirmation of the expression of M-CSFR in T47D *FMS* but not T47D cells was performed by flow cytometry (Figure 3A).

To compare CAR function, T-cells that expressed this panel of CARs and truncated controls were co-cultivated overnight with T47D and T47D *FMS* tumour cells. We found that all CD3ζ-containing CARs mediated efficient killing of T47D *FMS* but not parental T47D cells (Figure 3B), thereby confirming the specificity of these CARs for M-CSFR. As expected, no cytotoxic activity was observed with CD3ζ-deficient endodomain truncated controls (M-Tr and 34-Tr; Figure 3B). Supernatant was collected from these co-cultures and analysed for T-cell-derived cytokine production by ELISA. Notably, production of both IFN-γ (Figure 3C) and IL-2 (Figure 3D) was significantly increased by M-2 and M-3 CAR T-cells, when compared to 34-2 and 34-3 T-cells. In addition, cytokine release trended (IL-2; Figure 3D) or was significantly higher (IFN-γ; Figure 3C) by M-2 than M-3 T-cells.

### 3.3. M-CSF-Targeted CAR T-Cells Mediate Enhanced Cytolytic Activity against M-CSFR^+^ Lymphoma Cells

While the experiments described above confirm the target specificity of these CARs, a limitation arises from the fact that artificially high M-CSFR expression is achieved following retroviral engineering of T47D cells with SFG *FMS*. We next set out to address whether these CARs could confer specificity against target cells that naturally express M-CSFR. We focused on Hodgkin’s disease (HD) and anaplastic large-cell lymphoma (ALCL) since derived cell lines express readily detectable cell surface M-CSFR [29,30]. In agreement with previous reports, we demonstrated that these cells contain a range of levels of M-CSFR by flow cytometry (Figure 4A). Cell surface expression of M-CSFR could not be detected using this assay in monocytes (Mo), although M1- and M2-polarised macrophages were both positive for M-CSFR expression (Figure 4A). All these lymphoma cell lines produced either M-CSF and/or IL-34, unlike either T47D or T47D *FMS* cells (Figure 4B). Conceivably, this could hinder tumour cell targeting as a result of M-CSFR blockade by soluble ligand, which could compete with the CAR targeting moiety. Such an immune evasion mechanism has recently been described for CAR T-cells targeted against glypican-3 [31].

To compare CAR function in these more stringent models, lymphoma cells were engineered to co-express firefly luciferase and tandem dimer tdTomato red fluorescent protein (ffLuc/RFP), and then flow sorted to purity (Appendix A). M-CSF- and IL-34-targeted CAR T-cells were co-incubated with these engineered lymphoma cells and the viability of the latter was monitored daily for six days using luciferase assays. A comparison was made with lymphoma cells that were cultured alone (set to 100% tumour cell viability). This analysis revealed that all CD3ζ-containing CARs mediated tumour cell killing (Figure 4C) over the six day co-cultivation assay. M-CSF-targeted CARs significantly outperformed their IL-34-containing counterparts. This was most notable when 2G CARs were compared, indicated by the significantly greater cytolytic activity of M-2 against all six lymphoma cell lines, when compared to 34-2 (Figure 4C). Across this panel of lymphoma cells, no consistent differences were evident between 2G and 3G CARs in which targeting was achieved by either ligand (Figure 4C). Lymphoma cell cytotoxicity correlated with the intensity of cell surface M-CSFR expression, although this correlation was stronger for M-CSF- compared to IL-34-containing CARs (Appendix A). Notably, some tumour cell lines with similar levels of M-CSFR expression (e.g., KM-H2 and L540) exhibited differing sensitivities to killing by CAR T-cells. This may be due to differences in the intrinsic susceptibility of these two cell lines to undergo apoptosis upon encounter with activated CAR T-cells. Moreover, autocrine release of IL-34 (the high-affinity ligand for M-CSFR) was greater by L540 cells, and this may have compromised CAR T-cell targeting with greater efficiency than was achieved by M-CSF release from KM-H2 cells.

### 3.4. M-CSF-Containing CAR T-Cells Produce Higher Cytokine Levels When Cultured with Lymphoma Cells That Express M-CSFR

Next, we compared cytokine production by these CAR T-cells when co-incubated with the six M-CSFR-expressing lymphoma cell lines described above. Despite considerable donor to donor variation, we observed that M-CSF-containing CAR T-cells produced greater amounts of both IFN-γ (Figure 5A) and IL-2 (Figure 5B) compared to IL-34-targeted CAR T-cells. Once again, no differences between 2G and 3G CAR T-cells were observed in these experiments. When cytokine production data were pooled for all six tumour cell lines, significantly greater IFN-γ and IL-2 production was observed for both 2G and 3G CARs in which M-CSF rather than IL-34 was used to confer specificity (Appendix A).

## 4. Discussion

While CAR specificity is most commonly conferred by an scFv, alternative targeting options include the use of peptide ligands [18], cytokines [32], chimeric cytokine derivatives [20] or natural receptors [33]. In this study, we confirmed the feasibility of engineering M-CSFR-specific 2G and 3G CAR T-cells that employ either M-CSF or IL-34 to confer targeting specificity. Importantly, these CARs can mediate the recognition of lymphoma cells that naturally express low levels of M-CSFR, despite their autocrine secretion of potentially competitive M-CSF and/or IL-34 ligands. Notably, we found that the choice of targeting moiety exerted a strong influence on the functionality of these receptors, with M-CSF-containing CARs outperforming their IL-34-containing counterparts. Since M-CSF and IL-34 bind competitively to the same region of the M-CSFR, this functional difference is not attributable to the need for individually optimised CAR spacer domains, as is required for epitopes that sit at different distances from the tumour cell membrane [34]. One possible explanation for this finding relates to the divergent affinity of these two targeting moieties for M-CSFR. Most chimeric antigen receptors contain targeting moieties with nanomolar range affinity. In contrast, IL-34 has a K_d_ of 1 pM owing to an extremely fast on-rate and a slower off-rate for M-CSFR, when compared to M-CSF [15]. Accordingly, IL-34-targeted CARs have the highest affinity that has been tested to date, and our data suggest that this extreme affinity may be detrimental to CAR T-cell function.

It is well-known that excessive affinity can compromise the ability of T-cell receptor engagement to trigger T-cell activation [35,36,37]. However, prior investigations of the importance of affinity in CAR function have yielded mixed findings. Some investigations have shown that high-affinity CARs maintain satisfactory anti-tumour activity, whereas their lower-affinity counterparts selectively kill tumour cells that express elevated target antigen levels while sparing normal cells in which the target is found at lower levels [38,39,40,41]. In these publications, the maximum scFv affinity evaluated was K_d_ = 1.5 × 10^−11^ M, which is 15-fold lower than that of IL-34 for M-CSFR. Other groups have reported that high-affinity CARs outperform lower-affinity variants [42,43]. Once again, however, scFvs used in these studies have nanomolar affinities that are much below that of IL-34 for M-CSFR. On the other hand, some nanomolar affinity CARs have been shown to sequester the target ligand, potentially hindering their ability to undertake serial tumour cell killing [44]. In keeping with this, micromolar affinity CARs with specificity for ICAM-1 or CD38 outperformed their nanomolar counterparts in discriminating between normal and transformed cells, while maintaining strong efficacy against tumour cells [45,46]. Many factors are likely to contribute to these different findings, including differences in CAR architectures, target antigens and both epitope location and density. Nonetheless, recent clinical evidence also provides support for the use of lower-affinity CAR-targeting moieties. Patients with relapsed refractory B-cell acute lymphoblastic leukaemia were treated with CD19-specific CAR T-cells that had a greater than 40-fold reduction in affinity compared to gold standard FMC63 scFv-containing CARs. T-cells that expressed the lower-affinity CAR demonstrated improved performance in pre-clinical testing compared to a matched FMC63 CAR and maintained excellent clinical efficacy, but without any evidence of severe toxicity [47]. It should also be noted in this context that FMC has low nanomolar affinity for CD19 (K_d_ 3 × 10^−9^ nM) [48].

Affinity is determined by both on-rate and off-rate, each of which may be important in determining the anti-tumour function. It has been suggested that a reduced on-rate may better enable CAR T-cells to distinguish between high-antigen expressing tumour cells and healthy cells in which lower levels of antigen are present [40]. On the other hand, an increased off-rate may facilitate the serial killing function [49] and reduce both exhaustion and activation-induced cell death [45], all of which are desirable attributes of tumour-specific T-cells. These considerations provide additional conceptual grounds to speculate that excessive CAR affinity would be disadvantageous for anti-tumour activity.

A further significant finding of our study was the lack of superior performance of 3G CARs targeted against M-CSFR, when compared to CD28-containing 2G designs. While some 3G CARs have outperformed matched 2G controls, [12,50,51,52,53,54], others demonstrate marginal superiority [55,56,57], or inferior anti-tumour function [58,59]. We have recently shown that co-stimulation requires membrane proximity, a property that cannot be achieved for both co-stimulatory units within a linear 3G CAR. For this reason, we have engineered parallel CAR systems in which both co-stimulatory units occupy a membrane proximal position, a finding that we have linked to superior anti-tumour activity [60].

Further development of the CARs described in this manuscript will require testing of their anti-tumour activity against primary tumour cells and in vivo models. It is also acknowledged that our study has a number of limitations. First, cytotoxicity assays were conducted at a 1:1 effector to target ratio, which is unlikely to be attainable in the tumour microenvironment. Moreover, experiments which test a greater range of effector to target ratios may uncover further differences between the function of the CARs under study. Analysis of exhaustion marker expression in co-cultivation experiments could provide additional mechanistic insight into the functional differences between CARs described above. There is also a need for further investigation of whether autocrine release of ligands of the M-CSFR can inhibit effective CAR T-cell-mediated targeting of this receptor on tumour cells. Finally, assessment of the interaction between these CARs and tumour-associated macrophages also warrants investigation.

## 5. Conclusions

We described a panel of 2G and 3G CARs with specificity for M-CSFR and showed that they mediate target-dependent T-cell activation and tumour cell killing. Despite sharing overlapping epitopes within M-CSFR, low picomolar affinity CARs in which targeting is achieved by IL-34 performed worse than those containing M-CSF, which had a 34-fold lower affinity for M-CSFR. These data suggest that extreme affinity impairs CAR T-cell function, in contrast to the desirability of this attribute in monoclonal antibodies.

## Figures and Tables

**Figure 1 cells-11-02190-f001:**
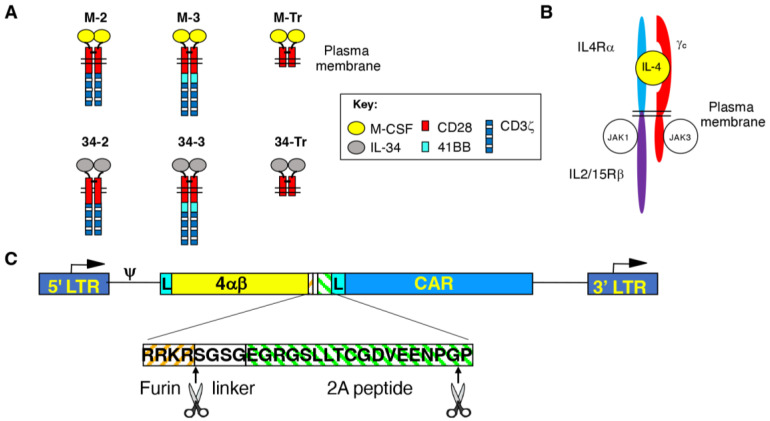
Vector and chimeric antigen receptor design. (**A**) Line diagrams depicting M-CSFR-specific 2G and 3G CARs and truncated (Tr) controls. Targeting specificity was achieved using M-CSF or IL-34, which are competing ligands of the M-CSFR. CARs are presumed dimers owing to the dimeric nature of M-CSF and IL-34 and the presence of a cysteine in the CD28 spacer that promotes disulphide bond formation. (**B**) Cartoon structure of the 4αβ chimeric cytokine receptor. 4αβ contains the ectodomain of IL-4 receptor α joined to the transmembrane and endodomain of the IL-2/15 receptor β. Upon binding of IL-4, 4αβ heterodimerises with the common γ chain, thereby delivering a potent mitogenic signal to the gene-modified T-cells. (**C**) Schematic illustration of the SFG retroviral vectors used to co-express CARs and 4αβ. The vector long-terminal repeats (LTR), encoded leader peptides (L) and an intervening furin cleavage site/linker/T2A ribosomal skip sequence between 4αβ and CAR cDNAs are indicated.

**Figure 2 cells-11-02190-f002:**
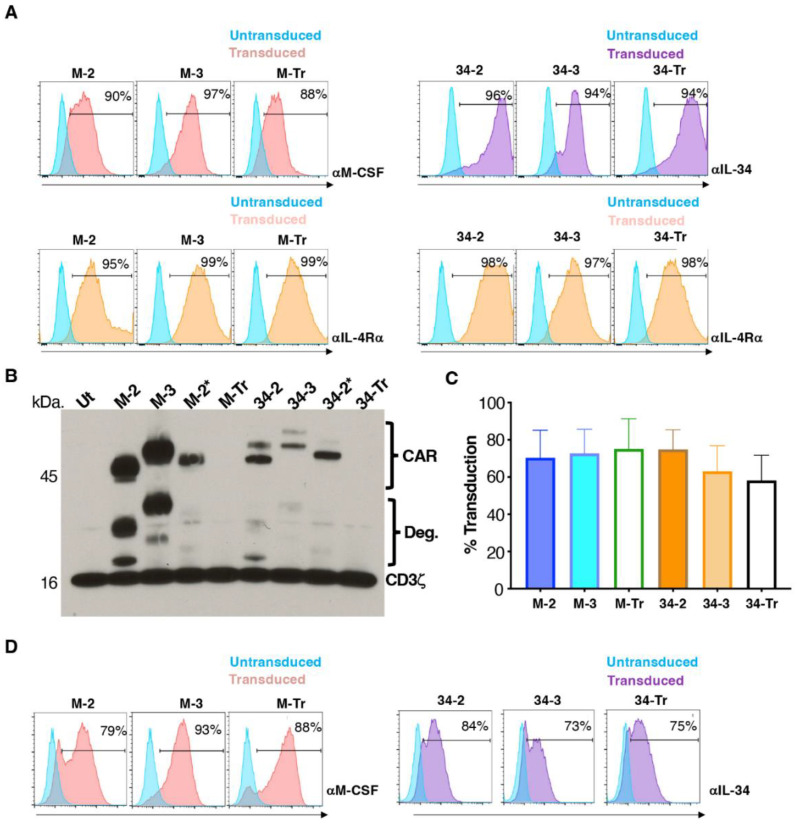
Engineering of human T-cells to express M-CSFR-specific CARs. (**A**) Cell surface expression of CARs and 4αβ receptor on 293VEC-RD114^TM^ retroviral packaging cells. Similar findings were obtained using PG13 retroviral packaging cells. (**B**) A Western blot was prepared under reducing conditions using protein lysates derived from the indicated CAR T-cell populations. Following incubation with an anti-CD3ζ antibody, endogenous CD3ζ chain and CAR-associated bands are highlighted. Endodomain truncated (Tr) CARs serve as negative controls. Deg. indicates presumed CAR degradation products. M-2* and 34-2* contain a T2A peptide fused to the CAR C-terminus, in contrast to all other CARs. (**C**) Cell surface levels of the indicated CARs in retrovirus transduced human T-cells following IL-4-mediated enrichment (mean ± SD, n = 10). (**D**) Representative examples of expression of M-CSFR-specific 2G, 3G and truncated CARs in human T-cells. Analysis was performed by flow cytometry.

**Figure 3 cells-11-02190-f003:**
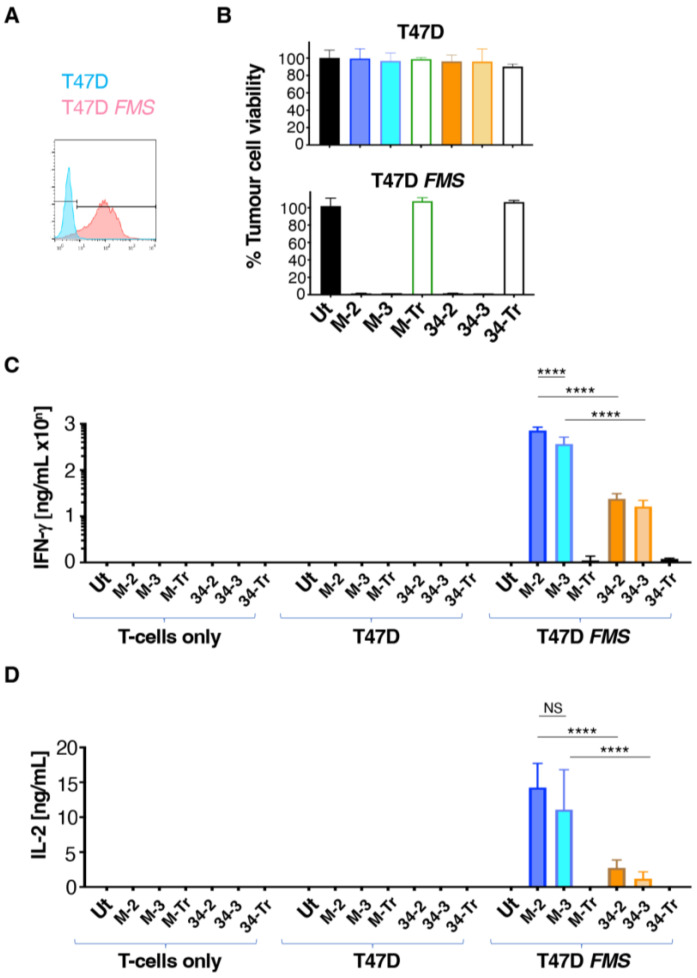
Co-cultivation of CAR T-cells with T47D tumour cells discordant for M-CSFR expression. (**A**) Flow cytometric analysis of cell surface M-CSFR expression in T47D and T47D *FMS* cells. (**B**) Engineered human T-cells were co-cultivated overnight at a 1:1 effector:target ratio with T47D (lacks M-CSFR) or T47D *FMS* (expresses M-CSFR) target cells. Tumour cell viability was evaluated using an MTT assay (mean ± SD, n = 3 technical replicates). Data are representative of 7 independent replicates, in which similar results were obtained. Supernatant collected at termination of these cultures was analysed for IFN-γ (**C**) and IL-2 (**D**) concentrations (mean ± SD, n = 3 biological replicates). Statistical analysis was undertaken using two-way ANOVA and selected differences are shown. **** *p* < 0.0001; NS–not significant.

**Figure 4 cells-11-02190-f004:**
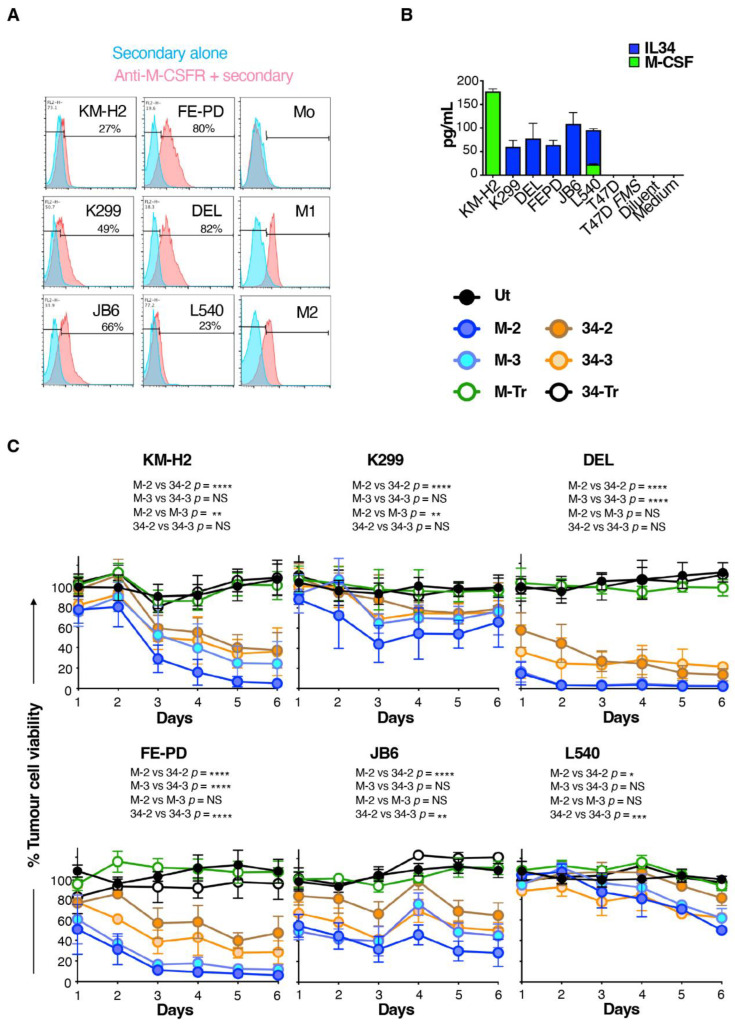
Lymphoma cell killing activity of M-CSFR-specific CAR T-cells. (**A**) Flow cytometry analysis of cell surface M-CSFR expression by the specified lymphoma cell lines, making comparison with monocytes (Mo) and M1- or M2-polarised macrophages as controls. (**B**) ELISA analysis of conditioned medium collected from the specified lymphoma cell lines for IL-34 and M-CSF (mean ± SD, n = 3). (**C**) T-cells were engineered to express the specified CARs and then co-cultivated at a 1:1 ratio with the indicated lymphoma cell lines. Lymphoma cell viability was determined each day using a luciferase assay (mean ± SD, n = 5) and was normalised to lymphoma cells cultured without T-cell addition (set to 100%). Two-way ANOVA was performed (comparing tumour cell viability over days 1–6). Selected significant differences are indicated. **** *p* < 0.0001; *** *p* < 0.001; ** *p* < 0.01; * *p* < 0.05.

**Figure 5 cells-11-02190-f005:**
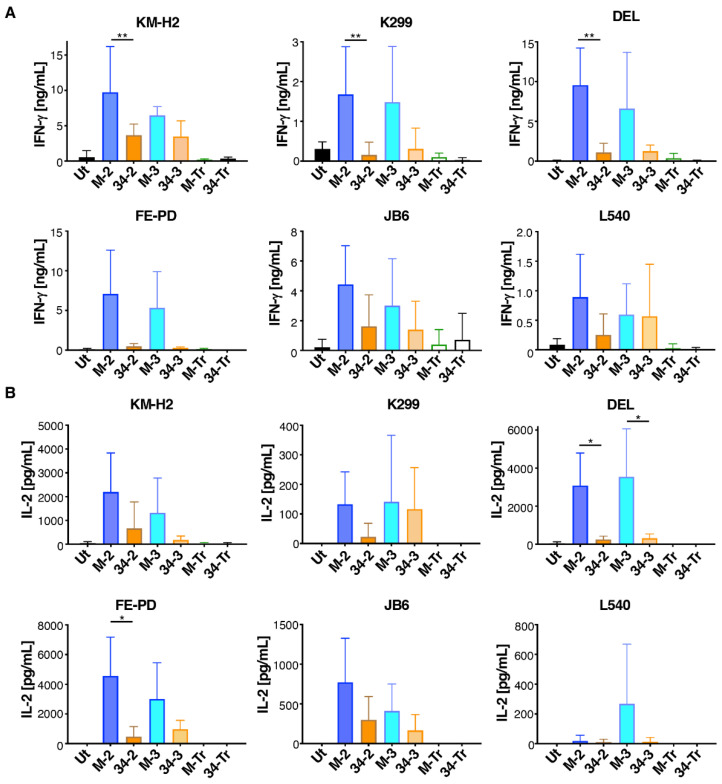
Cytokine production by M-CSFR re-targeted T-cells when co-cultured with lymphoma cell lines. T-cells were engineered to express the specified CARs and then co-incubated at a 1:1 effector:target ratio with the indicated ffLuc/RFP^+^ lymphoma cell lines. Interferon-γ (mean ± SD, n = 3–6 independent biological replicates (**A**)) and interleukin 2 (mean ± SD, n = 3–4 independent biological replicates (**B**)) were measured in supernatants harvested after 24 h. One-way ANOVA was undertaken and selected significant differences are indicated. ** *p* < 0.01; * *p* < 0.05.

## Data Availability

Data are available from the corresponding author upon request.

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
