# Peer review of "CAR T-Cell Targeting of Macrophage Colony-Stimulating Factor Receptor"

_cells, 2022, doi:10.3390/cells11142190_

Round 1

Reviewer 1 Report

In this manuscript, authors presented constructs of CAR-T-cell targeting M-CSFR expressing cells.  They characterized them by in vitro evaluating cytolyze of M-CSFR expressing cells.

Overall, experiments were well performed. However, they need complementary information and experiments to render this manuscript more convincing.

1.      As mentioned by the authors, the question of the expression level of M-CSFR is important. However, why M-CSFR receptor was not quantified in cell lines tested ? Several kits exist now and they could provide data to support a quantitative comparison. That could also help to establish whether there is a correlation between cytotoxicity and M-CSFR expression. The correlation mentioned by the authors is not obvious. How we can explain that, based on Fig4A, M-CSFR expression in KM-H2  and L540 seems to be similar whilst the sensitivity to CAR-T cells is so different ? To establish (or not) such a correlation and to discuss more in details this point, the best way must be to plot the IC50 vs the quantification of M-CSFR.  

2.      Minor point: In Fig3B and 4C, please, use the term “(tumour) cell viability” and not tumour viability, this one is more appropriate for in vivo experiment. In the text, lymphoma cells is also more appropriate than tumour cells for all cell lines derived from lymphoma.

3.      Concerning the viability experiments, that needs complementary information and experiment as well. Incubation time was not specified for experiment shown in Fig3B. In viability assays, only a co-culture ratio of 1:1 was tested. Indeed, this ratio does not represent “the real life” and a ratio close to expected in clinic should be tested to really evaluate the potency of such therapy (this point was never discussed by the authors).

4.      Discussion about the comparison of efficacy between 2G and 3G CAR-T cells should be enriched by experiments in more competitive conditions for binding. As mention above, another ratio could lead to a more competitive situation and differences between constructs of CAR-T cells could appear.  The question of competition with natural ligands was only slightly evoked by the Fig4B, commented in one sentence and not a really discussed. Is the viability assay sensitive to the ligand produced by lymphoma cells ? Yes ? No ? Why ? By performing experiment with various concentrations of competitive ligands, a difference between the 2G and 3G CAR-T cells could be easier to observed.

Author Response

Reviewer 1

In this manuscript, authors presented constructs of CAR-T-cell targeting M-CSFR expressing cells.  They characterized them by in vitro evaluating cytolyze of M-CSFR expressing cells.

Overall, experiments were well performed. However, they need complementary information and experiments to render this manuscript more convincing.

  1. As mentioned by the authors, the question of the expression level of M-CSFR is important. However, why M-CSFR receptor was not quantified in cell lines tested? Several kits exist now and they could provide data to support a quantitative comparison. That could also help to establish whether there is a correlation between cytotoxicity and M-CSFR expression. The correlation mentioned by the authors is not obvious. How we can explain that, based on Fig4A, M-CSFR expression in KM-H2 and L540 seems to be similar whilst the sensitivity to CAR-T cells is so different ? To establish (or not) such a correlation and to discuss more in details this point, the best way must be to plot the IC50 vs the quantification of M-CSFR.  

Response: We apologise that quantification of M-CSFR was omitted from FACS plots shown in Figure 4A of our manuscript. These have now been added. We did find a correlation between percentage M-CSFR expression and lymphoma cell cytotoxicity by each active CAR, as shown in Supplementary Figure 2. Unfortunately however, killing assays have not been performed at a range of effector to target ratios to allow us to plot IC50. The reviewer is correct to point out the difference in sensitivity of the KM-H2 and L540 cell lines to CAR T-cell mediated killing. This is not an uncommon finding in CAR T-cell killing assays and may relate to the intrinsic susceptibility of the two cell lines to undergo apoptosis upon encounter with activated CAR T-cells. Moreover, autocrine release of IL-34 (the high affinity ligand for M-CSFR) was greater by L540 cells and this may have compromised effective CAR T-cell targeting more efficiently than was achieved by M-CSF release from KM-H2 cells. To address this point, a sentence has been added to read:

“Notably, some tumour cell lines with similar levels of M-CSFR expression (e.g. KM-H2 and L540) exhibited differing sensitivities to killing by CAR T-cells. This may be due to differences in the intrinsic susceptibility of these two cell lines to undergo apoptosis upon encounter with activated CAR T-cells. Moreover, autocrine release of IL-34 (the high affinity ligand for M-CSFR) was greater by L540 cells and this may have compromised CAR T-cell targeting with greater efficiently than was achieved by M-CSF release from KM-H2 cells.”

  1. Minor point: In Fig3B and 4C, please, use the term “(tumour) cell viability” and not tumour viability, this one is more appropriate for in vivo experiment. In the text, lymphoma cells is also more appropriate than tumour cells for all cell lines derived from lymphoma.

Response: These changes have been made.

  1. Concerning the viability experiments, that needs complementary information and experiment as well. Incubation time was not specified for experiment shown in Fig3B. In viability assays, only a co-culture ratio of 1:1 was tested. Indeed, this ratio does not represent “the real life” and a ratio close to expected in clinic should be tested to really evaluate the potency of such therapy (this point was never discussed by the authors).

Response: In Fig 3B, the co-culture was performed overnight prior to measurement of tumour cell viability. To indicate this, the Figure legend reads: “Engineered human T-cells were co-cultivated overnight (1:1 ratio) with T47D (lacks M-CSFR) or T47D FMS (expresses M-CSFR) target cells. Residual tumor cell viability was determined using an MTT assay (mean + SD, n=3 technical replicates).” The author is correct to point out that an E:T ratio of 1:1 is unlikely to be attained within a tumour. However, our primary focus was to compare activity of M-CSF and IL-34-containing CARs using a range of tumour and lymphoma target cells rather than to simulate clinical application of these CARs. To acknowledge this point, a sentence has been added to the Discussion which reads: “Our study has a number of limitations. First, cytotoxicity assays were conducted at a 1:1 effector to target ratio which is unlikely to be attainable in the tumour microenvironment.”  

  1. Discussion about the comparison of efficacy between 2G and 3G CAR-T cells should be enriched by experiments in more competitive conditions for binding. As mention above, another ratio could lead to a more competitive situation and differences between constructs of CAR-T cells could appear.  The question of competition with natural ligands was only slightly evoked by the Fig4B, commented in one sentence and not a really discussed. Is the viability assay sensitive to the ligand produced by lymphoma cells ? Yes ? No ? Why ? By performing experiment with various concentrations of competitive ligands, a difference between the 2G and 3G CAR-T cells could be easier to observed.

Response: We agree with the reviewer that these would represent insightful additional experiments although we would respectfully suggest that this is beyond the scope of the current manuscript. To acknowledge this point, a final paragraph has been added to the Discussion which reads: “Our study has a number of limitations. First, cytotoxicity assays were conducted at a 1:1 effector to target ratio which is unlikely to be attainable in the tumour microenvironment. Experiments which test a greater range of effector to target ratios may uncover further differences between the function of the CARs under study. There is also a need for further investigation of whether autocrine release of ligands of the M-CSFR can inhibit effective CAR T-cell mediated targeting of this receptor on tumour cells.”

Reviewer 2 Report

The authors elegantly describe the process of CAR T-cell targeting of macrophage colony-stimulating factor receptor. The paper is well written and the methods and figures are clear. 

Major Comments:

1. The authors clearly describe a translational research problem in the introduction section. However, the authors only focus on the generation of CARs in in vitro experiments, without addressing the use of these CAR T cells in any kind of translational setting. It would be of high interest to also include some ex vivo experiments to show that the generated CAR T cells can also kill human primary tumor cells (instead of cell lines).

2. M-CSF and M-CSFR are important factors in macrophage influx in tumors. However, the authors do not address the impact of CAR T-cell targeting of M-CSFR on macrophages. It would be of high value to perform some migration assays with macrophages in the presence and absence of CAR-T cells directed against M-CSFR. 

Minor Comments:

1. Please include color legends for Figure 2A (in a similar way as Figure 2C)

2. Figure 2B is unclear, both in the text (lines 202-208) and in the image itself. Please specify the predicted molecular weight for all different CARs, and explain all bands that are observed. 

3. The term tumour viability in Figure 3B and Figure 4Cis misleading as it suggests that the assay was performed on tumour tissue (not a cell line), furthermore it does not refer to the fact that a cytotoxicity assay was performed. Please rephrase (%lysis or %cytotoxicity)

4. Line 248: change greater to increased

5. In line 253, the authors refer to "these studies". Please specify to which studies. 

6. In the introduction, the authors mention the presence of M-CSFR in different cancer types. In their study, they only focus on some of these. Please briefly explain the reason behind focussing on Hodgkins disease/ lymphomas.

7. Please address the future plans for these CAR T cells in a translational setting in the discussion section. What are the next steps?

Author Response

Reviewer 2

Major Comments:

  1. The authors clearly describe a translational research problem in the introduction section. However, the authors only focus on the generation of CARs in in vitro experiments, without addressing the use of these CAR T cells in any kind of translational setting. It would be of high interest to also include some ex vivo experiments to show that the generated CAR T cells can also kill human primary tumor cells (instead of cell lines).

Response: We agree with the reviewer that testing of anti-tumour activity against primary human tumour cells and in vivo models would be necessary steps in the further development of the CARs described in this study. To acknowledge this point, we have added the following sentence to the Discussion: “Further development of the CARs described in this manuscript would require testing of their anti-tumour activity against primary tumour cells and in vivo models.”  

  1. M-CSF and M-CSFR are important factors in macrophage influx in tumors. However, the authors do not address the impact of CAR T-cell targeting of M-CSFR on macrophages. It would be of high value to perform some migration assays with macrophages in the presence and absence of CAR-T cells directed against M-CSFR. 

Response: Once again, the reviewer makes an excellent suggestion for further experimental work. We have acknowledged this point by adding the following sentence to the Discussion: “….assessment of the interaction between these CARs and tumour-associated macrophages also warrants investigation.”

 Minor Comments:

  1. Please include color legends for Figure 2A (in a similar way as Figure 2C)

Response: This has been added.

  1. Figure 2B is unclear, both in the text (lines 202-208) and in the image itself. Please specify the predicted molecular weight for all different CARs, and explain all bands that are observed.

 Response: We have sought to clarify the description of the western blot in a revised paragraph, which reads: “Following transduction of human T-cells, western blotting analysis was performed under reducing conditions to investigate if CARs expressed at the predicted molecular weights (mw) of 45.6 kDa (M-2), 50.5 kDa (M-3), 52.9 kDa (34-2) and 58.2 kDa (34-3) (Figure 2B). The blot was probed with anti-CD3z antibody which allows detection of CAR band(s) together with endogenous CD3z. CAR and presumed degradation (Deg.) bands are indicated on the blot. Duplicate lanes are shown for M-2 and 34-2 (second lane indicated by an asterisk). In the case of M-2* and 34-2*, CARs were fused to a C-terminal T2A peptide which accounts for the slightly increased molecular mass in each case. Two CAR bands were noted in the case of IL-34 CARs, perhaps owing to its propensity to undergo C-terminal glycosylation [ref]. The endogenous CD3z band was seen at similar intensity in all lanes, conforming equal protein loading of each lane.”

  1. The term tumour viability in Figure 3B and Figure 4C is misleading as it suggests that the assay was performed on tumour tissue (not a cell line), furthermore it does not refer to the fact that a cytotoxicity assay was performed. Please rephrase (%lysis or %cytotoxicity)

Response: We have changed this to read % tumour cell viability, also in line with the comment raised by reviewer 1.

  1. Line 248: change greater to increased

Response: This has been changed.

  1. In line 253, the authors refer to "these studies". Please specify to which studies. 

Response: This sentence has been changed to read: “While the experiments described above confirm target specificity of these CARs, …”

  1. In the introduction, the authors mention the presence of M-CSFR in different cancer types. In their study, they only focus on some of these. Please briefly explain the reason behind focussing on Hodgkins disease/ lymphomas.

Response: We have added an explanation in section 3.3 where work using lymphoma cell lines is introduced. Our focus on these disease types was simply because of availability of derived tumour cell lines that naturally express cell surface M-CSFR. Despite the literature that describes aberrant expression of M-CSFR in a range of solid tumours, we have not found that derived cell lines express detectable M-CSFR on the cell surface. The following sentence has been added: “We focused on Hodgkin’s disease (HD) and anaplastic large cell lymphoma (ALCL) since derived cell lines express readily detectable cell surface M-CSFR.”

  1. Please address the future plans for these CAR T cells in a translational setting in the discussion section. What are the next steps?

Response: We have added a sentence to the Discussion to outline next steps for these CARs. This reads: “Further development of the CARs described in this manuscript will require testing of anti-tumour activity against primary tumour cells and in vivo models.”

Reviewer 3 Report

The manuscript by Achkova et al. reported the development of CARs targeting M-CSFR using its natural ligands M-CSF or IL-34. The authors demonstrated that both 2nd and 3rd generation versions of M-CSF and IL-34 containing CARs have selective cytotoxicity against M-CSFR expressing cell lines (engineered or endogenous expression), accompanied by IL-2 and IFN-gamma secretion, with M-CSF CAR outperforming IL-34 CAR in cytotoxicity and cytokine secretion.

The manuscript is well written and the conclusions in the manuscript are well supported by the data. The development of natural ligand-based M-CSFR CARs can provide alternatives for current scFv-based CAR T therapies. Comparison of low and high affinity ligands also provide new insights on general CAR design strategies. Overall, the results are convincing with control experiments and the statistics were carefully done. I support the publication of this work after addressing the concerns below.

Major:

Since the motivation of the study to generate CARs with natural ligands is because “scFv-based CARs are susceptible to tonic signaling”, a comparison with scFv-based M-CSFR CARs in terms of tonic signaling should be performed. However, I did not see any characterization on tonic signaling for the new CARs reported.

The comparison between M-CSF and IL-34 CARs. Although both CARs demonstrated specific cytotoxicity against M-CSFR, does the low affinity M-CSF CAR have better discrimination between normal cells versus tumour cells? It would provide more insights on why M-CSF CAR can outperform IL-34 CAR if authors can compare the on- and off-rates between these two.

The cytotoxicity assays (figure 4). The co-culture time (6 days) is much longer than other reports (e.g., 18 hr in PMID:28225754). What is the motivation for the long-time assay? Also, since long time co-cultures were performed, it would be interesting to look at the exhaustion markers of different CAR T cells after the assays and compare the persistence.

Minor:

Section 3.3 is missing.

Line 95-96: it is not clear how the furin cleavage site was used in the study.

Line 263-264: Any observed evidence or reference for the effects of M-CSF and IL-34 on the CAR T performance?

Author Response

Reviewer 3

The manuscript by Achkova et al. reported the development of CARs targeting M-CSFR using its natural ligands M-CSF or IL-34. The authors demonstrated that both 2nd and 3rd generation versions of M-CSF and IL-34 containing CARs have selective cytotoxicity against M-CSFR expressing cell lines (engineered or endogenous expression), accompanied by IL-2 and IFN-gamma secretion, with M-CSF CAR outperforming IL-34 CAR in cytotoxicity and cytokine secretion.

The manuscript is well written and the conclusions in the manuscript are well supported by the data. The development of natural ligand-based M-CSFR CARs can provide alternatives for current scFv-based CAR T therapies. Comparison of low and high affinity ligands also provide new insights on general CAR design strategies. Overall, the results are convincing with control experiments and the statistics were carefully done. I support the publication of this work after addressing the concerns below.

Major:

Since the motivation of the study to generate CARs with natural ligands is because “scFv-based CARs are susceptible to tonic signaling”, a comparison with scFv-based M-CSFR CARs in terms of tonic signaling should be performed. However, I did not see any characterization on tonic signaling for the new CARs reported.

Response: The reviewer is correct in highlighting the issue of tonic signalling with many scFv based CARs. Unfortunately however, we do not have access to scFv(s) with specificity for M-CSFR in order to make this comparison. We do not see evidence of tonic signalling by the CARs described in our report, indicated by the lack of cytokine production by these CAR T-cells when co-cultivated with target cells that lack M-CSFR (e.g. Figure 3C-D). We have also toned down the reference to tonic signalling (change italicized) by stating: “Since some scFv-based CARs are susceptible to tonic signalling [ref], we set out to engineer 2G and 3G CARs in which specificity for M-CSFR is conferred by either of the two natural ligands of this receptor.”

The comparison between M-CSF and IL-34 CARs. Although both CARs demonstrated specific cytotoxicity against M-CSFR, does the low affinity M-CSF CAR have better discrimination between normal cells versus tumour cells? It would provide more insights on why M-CSF CAR can outperform IL-34 CAR if authors can compare the on- and off-rates between these two.

Response: We did not undertake toxicity testing of the CARs but would not anticipate an advantage for the M-CSF CAR in this respect, given the ability of M-CSF to stimulate the differentiation of monocytes and the readily detectable expression of M-CSFR on macrophages of both M1 and M2 types. For these reasons, it is expected that M-CSF containing CARs would recognize M-CSFR-expressing normal tissue.

On-rates and off-rates of M-CSF and IL-34 for M-CSFR have been published by Lin et al (2008) Science, 320, 807-811, and are as follows:

IL-34               Kon 6.29 x 107 M-1 s-1 (authors acknowledge that this may be an underestimate)

                        Koff 6.55 x 10-5 s-1

                        Kd 1.04 pM

M-CSF           Kon 1.77 x 107 M-1 s-1

                        Koff 6.03 x 10-4 s-1

                        Kd 34.1 pM

Thus, the greater affinity of IL-34 is contributed to by a faster on-rate and a slower off-rate for M-CSFR, when compared to M-CSF. We have modified the Discussion to reflect this point as follows:

“Most chimeric antigen receptors contain targeting moieties with nanomolar range affinity. By contrast, IL-34 has a Kd of 1pM owing to an extremely fast on-rate and a slower off-rate for M-CSFR, when compared to M-CSF (ref). Accordingly, IL-34-targeted CARs have the highest affinity that has been tested to date and our data suggest that this extreme affinity may be detrimental to CAR T-cell function.”

The cytotoxicity assays (figure 4). The co-culture time (6 days) is much longer than other reports (e.g., 18 hr in PMID:28225754). What is the motivation for the long-time assay? Also, since long time co-cultures were performed, it would be interesting to look at the exhaustion markers of different CAR T cells after the assays and compare the persistence.

Response: The cytotoxicity assays performed with T47D tumour cells were performed overnight. However, the reviewer is correct to point out the more prolonged nature of the killing assays performed using lymphoma target cells. Our intention in this regard was to simulate the sustained encounter between CAR T-cells and tumour cells, which can cause exhaustion. It should be noted that viability data are presented for each daily time point in these assays. We agree than an analysis of exhaustion markers would have added significant value to these data. To acknowledge this point, we have added the following sentence to the shortcomings section of the Discussion. “Analysis of exhaustion marker expression in co-cultivation experiments could provide additional mechanistic insight into functional differences between CARs described above.”  

Minor:

Section 3.3 is missing.

Response: This has been corrected.

Line 95-96: it is not clear how the furin cleavage site was used in the study.

Response: The relevant section of the methods has been expanded to provide an explanation as follows: “All CARs were stoichiometrically expressed in human T-cells using a T2A ribosomal skip peptide and RRKR furin cleavage site alongside a chimeric cytokine receptor designated 4ab (Figure 1B). The purpose of the furin cleavage site was to remove the peptide overhang left by the T2A peptide on the upstream polypeptide.”

Line 263-264: Any observed evidence or reference for the effects of M-CSF and IL-34 on the CAR T performance?

Response: Antigen shedding has recently been shown to inhibit CAR T-cell immunotherapy, highlighting a theoretical risk that release of M-CSF or IL-34 could impede the biological activity of these CARs. This is of particular note in regard to IL-34, given its extremely high affinity for M-CSFR. To acknowledge this point, we have added a sentence to section 3.3 as follows: ”Such an immune evasion mechanism has recently been described for CAR T-cells targeted against glypican-3 [ref].”

Round 2

Reviewer 1 Report

The authors replied by adding data or completing the text. The answers are globally scientifically satisfying. 

However, for an unknown reason Figures 2 to 4 were added in double in the second version received. This has to be corrected.

Reviewer 2 Report

The authors have addressed all the issues raised, and have adapted the manuscript accordingly.